# Dimensional hierarchy of higher-order topology in three-dimensional sonic crystals

Xiujuan Zhang[1,6], Bi-Ye Xie[1,6], Hong-Fei Wang[1,6], Xiangyuan Xu[1,2], Yuan Tian[1], Jian-Hua Jiang[3]*, Ming-Hui Lu ◉ [1,4,5]* & Yan-Feng Chen[1,5]*

Wave trapping and manipulation are at the heart of modern integrated photonics and acoustics. Grand challenges emerge on increasing the integration density and reducing the wave leakage/noises due to fabrication imperfections, especially for waveguides and cavities at subwavelength scales. The rising of robust wave dynamics based on topological mechanisms offers possible solutions. Ideally, in a three-dimensional (3D) topological integrated chip, there are coexisting robust two-dimensional (2D) interfaces, one-dimensional (1D) waveguides and zero-dimensional (0D) cavities. Here, we report the experimental discovery of such a dimensional hierarchy of the topologically-protected 2D surface states, 1D hinge states and 0D corner states in a single 3D system. Such an unprecedented phenomenon is triggered by the higher-order topology in simple-cubic sonic crystals and protected by the space group $P_{m\bar{3}m}$. Our study opens up a new regime for multidimensional wave trapping and manipulation at subwavelength scales, which may inspire future technology for integrated acoustics and photonics.

[1] National Laboratory of Solid State Microstructures, Department of Materials Science and Engineering, Nanjing University, Nanjing 210093, China. [2] Key Laboratory of Noise and Vibration Research, Institute of Acoustics, Chinese Academy of Sciences, Beijing 100190, China. [3] School of Physical Science and Technology, and Collaborative Innovation Center of Suzhou Nano Science and Technology, Soochow University, 1 Shizi Street, Suzhou 215006, China. [4] Jiangsu Key Laboratory of Artificial Functional Materials, Nanjing 210093, China. [5] Collaborative Innovation Center of Advanced Microstructures, Nanjing University, Nanjing 210093, China. [6] These authors contributed equally: Xiujuan Zhang, Bi-Ye Xie, Hong-Fei Wang. *email: jianhuajiang@suda.edu.cn; luminghui@nju.edu.cn; yfchen@nju.edu.cn

opological insulators (TIs) with unprecedented boundary states, going beyond the classification of states of matter by spontaneous symmetry breaking, have stimulated tremendous research interest in electronic[1,2], photonic[3–11], and phononic[12–27] materials. Higher-order topological insulators (HOTIs), a novel paradigm for topological materials, exhibit unconventional bulk-boundary correspondence that enables lower-dimensional topological boundary states[28–43]. HOTIs offer new routes toward designer materials that give access to boundary wave localization in a topologically robust way in multiple dimensions. Such topological boundary wave localization can lead to robust wave guiding[3–9,13,15–18,22], frequency-stable cavity[44], unprecedented wave propagation[23], and other novel concepts for a paradigm of topologically robust chips in photonics[7] and acoustics[19].

For a $m$-dimensional ($m$D) TI, one can define the codimension of the $n$D boundary states as $l = m - n$. Then a $l$th-order TI is defined as a TI with $l$-codimensional topological boundary states[45,46]. Previously, the experimental realizations of HOTIs were focused on 2D materials by utilizing quadrupole topological insulators[32–34,42] and 2D TIs with quantized Wannier centers[37,39–41,43]. Theoretical predictions of 3D HOTIs have been proposed recently[28–31,35,38]. However, the experimental realization of 3D HOTIs[47] and the verification of the coexistence of 2D surface states, 1D hinge states and 0D corner states (i.e., HOTIs with 1, 2, 3-codimensional topological boundary states), still remain challenging.

Sonic crystals (SCs), a type of acoustic metamaterials with band structures that can be designed and tuned freely, have provided an ideal platform for the investigation of diverse topological states of matter such as the quantum Hall states[13,17], quantum spin Hall states[12,16], valley Hall states[18,22], topological crystalline insulators[39], and states with Weyl points and Fermi arcs[15,20,21,23,25,26]. Moreover, acoustic topological states in SCs can offer novel mechanisms for achieving acoustic cloaking[27], disorder-immune wave guiding[13,15–18,22], and topological negative refraction[23]. With versatile techniques for acoustic wave excitation and measurements, the search for novel topological states and phenomena in SCs has attracted lots of research interest[19]. Recently, such an endeavor is further facilitated by the 3D-printing technology for the fabrication of SCs[20–26].

In this article, we design and fabricate a 3D SC with a large bulk band gap that gives rise to topological boundary states with codimensions one, two and three. We observe directly the emergence of 2D topological surface states, 1D topological hinge states and 0D topological corner states, manifesting a dimensional hierarchy of topological boundary states due to higher-order band topology. The acoustic HOTI is characterized by the nontrivial bulk polarizations and the quantization of the Wannier centers (3D Zak phases)[35,48]. The underlying physics mimics topological crystalline insulators where the mirror symmetries restrict the positions of the Wannier centers to the maximal Wyckoff positions[49,50].

## Results

**Lattice structure and the higher-order topology.** Our 3D SCs have a simple-cubic lattice geometry (space group $P_{m\bar{3}m}$, no. 221) with a lattice constant $a = 2$ cm. In the 3D-printing technology, photosensitive resin (serving as "hard walls" for acoustic waves) is used as the printing material to fabricate the SCs. We start with an undeformed structure composed of eight air cavities located at the positions $0.25a(\pm1, \pm1, \pm1)$ (the origin of the coordinate is set at the center of the unit cell). These cavities are connected to their nearest neighbors by air channels (Fig. 1a, left). We use the solid areas to represent the hard walls, and the empty spaces to

represent the air cavities and the air channels. In order to characterize the deformation of the SCs, we introduce the center-to-center distance for the air cavities along the link within the unit cell, as $d_{intra}$. For the undeformed lattice, $d_{intra} = 0.5a$. By either reducing $d_{intra}$ (i.e., $d_{intra} < 0.5a$, denoted as shrunken in Fig. 1a, middle) or increasing $d_{intra}$ (i.e., $d_{intra} > 0.5a$, denoted as expanded in Fig. 1a, right), one can control the topology of the acoustic bands (we focus on the first acoustic band in this work). These deformations essentially mimic the Su–Schrieffer–Heeger physics[51] in three-dimensions, and thus naturally introduce the 3D Zak phase[52,53] $(\theta_x, \theta_y, \theta_z)$ to be defined below. Accordingly, the shrunken structure is topologically trivial, while the expanded structure is topologically nontrivial (the undeformed structure is the topological transition point). We consider the trivial (shrunken) SC with $d_{intra} = 0.15a$ and the topological (expanded) SC with $d_{intra} = 0.85a$, whose band structures and wavefunctions (as well as the parity eigenvalues) at the high-symmetry points are shown in Fig. 1b, c, respectively (see Supplementary Note 1 for another design). A complete acoustic band gap between the first and second bands with a large bandgap-to-midgap ratio of 41% emerges for both SCs. Note that the trivial and topological SCs have identical acoustic band structures, since they can be transformed into each other by a translation of the unit cell center. A similar scenario is known in the Su–Schrieffer–Heeger model. The distinct band topology of the two SCs is reflected on the parity and mirror eigenvalues of the first acoustic band at the high-symmetry points in the first Brillouin zone (BZ) (see the insets of Fig. 1b, c). The parity (and mirror) eigenvalue switching between the $\Gamma$ point and the other high-symmetry points in the BZ indicates the nontrivial topology of the SC with the expanded structure.

The topological properties of the SCs are further characterized by the bulk polarizations and the quantization of the Wannier centers (similar quantities are studied in the recent work on second-order TIs in 2D systems[35,40,41]). The 3D bulk polarization is defined as follows

$$p_i = -\frac{1}{(2\pi)^3} \int_{BZ} d^3\boldsymbol{k} \, \text{Tr}[\hat{\mathcal{A}}_i], \quad i = x, y, z \qquad (1)$$

where $(\hat{\mathcal{A}}_i)_{mn}(\mathbf{k}) = i\langle u_m(\mathbf{k})|\partial_{k_i}|u_n(\mathbf{k})\rangle$, with $m, n$ running over all bands below the bandgap. $|u_m(\mathbf{k})\rangle$ is the periodic part of the Bloch wavefunction of the $m$th band for the acoustic pressure field ($\mathbf{k}$ is the wavevector). In this work, only the first acoustic band is relevant to the topology of the band gap (i.e., $m = n = 1$). The Wannier center (WC) is located at $(p_x, p_y, p_z)$. The 3D bulk polarization is simply related to the 3D Zak phase[52] via $\theta_i = 2\pi p_i$ for $i = x, y, z$. Due to the three mirror symmetries with respect to the three orthogonal reflection planes, the WC is pinned at the maximal Wyckoff positions of the unit cell (see Supplementary Fig. 2). When the WC is at the center of the unit cell, the SC is adiabatically connected to the atomic limit which is topologically trivial. In contrast, the WCs of the topological crystalline insulators are away from the center of the unit cell (they are also denoted as obstructed atomic insulators[50]). There are three possible cases with nontrivial band topology: (i) when the WC locates at the center of the surfaces of the cubic unit cell, the insulator exhibits the first-order topology with topological surface states; (ii) when the WC locates at the center of the hinges of the unit cell, it exhibits the second-order topology with both topological surface and hinge states; (iii) when the WC locates at the corner of the unit cell, it exhibits the third-order topology, giving rise to the concurrent topological surface, hinge and corner states (see Supplementary Note 2).

The bulk polarizations are related to the Wannier bands. For the simple-cubic SCs studied in this work, we focus on the $z$

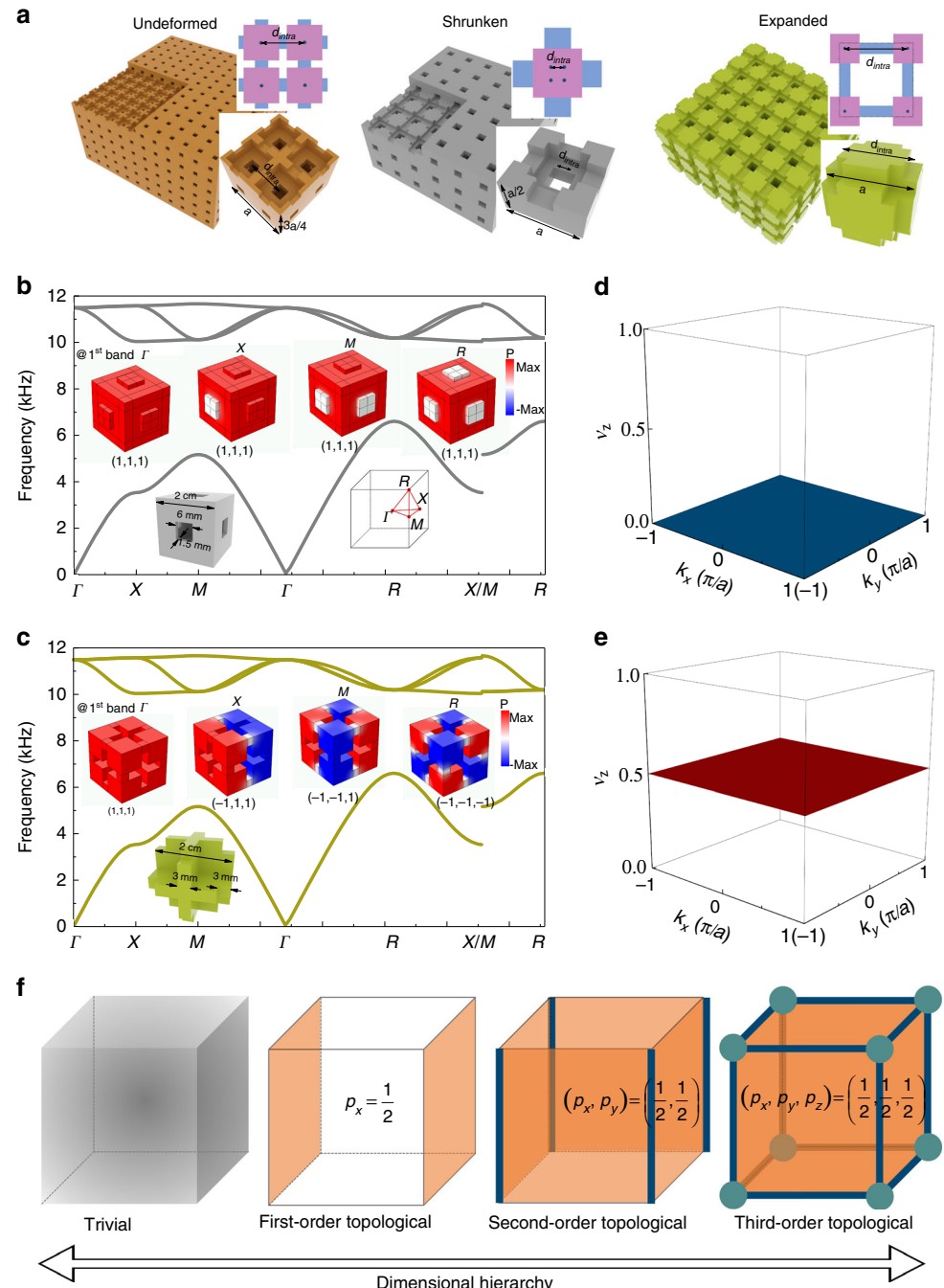

**Fig. 1** Dimensional hierarchy of the higher-order topology. **a** Schematics of the undeformed (left, gapless), shrunken (middle, trivial gap), and expanded (right, topological gap) lattices. The unit cell and its cross-section for each configuration are depicted in the insets. For the planar schematics in the insets, the purple regions denote the air cavities, while the blue regions denote the air channels. Calculated acoustic band structures for **b**, shrunken and **c**, expanded lattices are presented (the Brillouin zone and unit cells are shown in the insets). Acoustic wavefunctions at the high-symmetry points of the Brillouin zone and the mirror eigenvalues are also shown in the insets. **d**, **e** Wannier bands $\nu_z(k_x, k_y)$ for the trivial and topological sonic crystals (SCs), respectively. **f** Higher-order bulk-boundary correspondences as manifestation of the first-, second-, and third-order topology. The trivial phase with bulk states is also presented as comparison.

component of the bulk polarization (since $p_x = p_y = p_z$), $p_z = \frac{1}{(2\pi)^2} \int_S dk_x dk_y \nu_z(k_x, k_y)$, where $\nu_z(k_x, k_y)$ represents the Wannier band[43,48] and $S$ is the projection area of the BZ on the $k_x$-$k_y$ plane. We numerically calculate the Wannier band and find that for the trivial SC, $\nu_z = 0$ for all $(k_x, k_y)$ as shown in Fig. 1d, leading to $\boldsymbol{p} = (p_x, p_y, p_z) = (0, 0, 0)$. For the topological SC, however, the Wannier band takes a nontrivial and quantized value of $\nu_z(k_x, k_y) = 0.5$ as shown in Fig. 1e. Accordingly, the bulk polarization is also nontrivial, $\boldsymbol{p} = (\frac{1}{2}, \frac{1}{2}, \frac{1}{2})$, which indicates that the WC is located at the corner of the unit cell. Our SC is thus a HOTI of the case (iii). We would like to point out that the space group $P_{m\bar{3}m}$ consists of a minimum set of symmetries, including three mirror symmetries with respect to the $x = 0$, $y = 0$, and $z = 0$ planes as well as the $C_3$ rotation symmetry along the [111] direction. Such symmetry constraints allow only two

possible gapped phases: the trivial phase with $\boldsymbol{p} = (0, 0, 0)$ and the topological phase with $\boldsymbol{p} = (\frac{1}{2}, \frac{1}{2}, \frac{1}{2})$. These two gapped phases cannot be deformed into one another by continuous deformation of the SC without closing the bandgap or breaking the symmetries, thus forming the picture of symmetry-protected higher-order band topology. Based on the above analyses, such a HOTI will exhibit a dimensional hierarchy of the topological surface states (codimension 1), the topological hinge states (codimension 2), and the topological corner states (codimension 3). Figure 1f illustrates the emergence of these boundary states when the first-, second-, and third-order topological phases are manifested (a trivial phase with bulk states is also presented as comparison). It is seen that the surface of the second-order TI is a first-order TI which leads to the topological hinge states. The surface of the third-order TI is a second-order TI, which leads to the topological hinge and corner states. Such a scenario illustrates the dimensional hierarchy in our HOTI.

**Hierarchical topological boundary states in multi-dimensions.** To investigate the topological surface states, we numerically calculate the band structure (in the $k_x$–$k_y$ plane) for a ribbon-like supercell in Fig. 2a (the supercell is sketched in the inset). In-gap states (the orange curves) separated from the bulk bands and localized at the surface of the HOTI are found, showing the emergence of the topological surface states (see Supplementary Fig. 3 for their acoustic pressure profiles). We fabricate a "surface sample" (as shown in Fig. 2b) where the topological SC with $10 \times 10 \times 7$ unit cells is connected to the trivial SC of the same size. We perform three types of pump–probe measurements to detect and distinguish the bulk and surface states (see the

Methods section for details of the experiments; see the inset of Fig. 2c for the locations of the source and detector). The measured transmission spectra for the bulk and surface probes are presented in Fig. 2c. It is clearly seen that the surface probe captures most acoustic energy in the frequency region of 4.0–7.5 kHz. In this region, the excitation is dominated by the surface states, whereas the bulk state excitation is suppressed. Due to the finite-size effect, the transmission spectra of the bulk and surface deviate slightly from the eigenstates calculation (small fabrication errors ±0.1 mm may also contribute to such deviations; see the Methods section).

The measurement of the acoustic pressure profiles for the surface state excited by the source is presented using the slice plot in Fig. 2d. The measurement is performed using the same set-up as that in Fig. 2c, but we fix the frequency (5.5 kHz) and scan the pressure-field amplitudes with a spatial resolution of 2 mm along the $z$-direction and 2 cm along the $x$- and $y$-direction. The frequency 5.5 kHz is chosen because it corresponds to an obvious excitation peak of the surface probe, as well as the explicit suppression of the bulk probe. We observe a rapid decay of the acoustic energy away from the surface of the topological SC. Along the surface, the experimental data show that the acoustic wave can propagate, indicating topological wave localization on the surface. These results demonstrate that the surface and bulk states are distinguishable in the acoustic pump–probe measurements, even though the surface and bulk states are not spectrally well-separated (more evidence can be found in the Supplementary Note 4).

We now study the topological surface and hinge states by fabricating a "hinge sample" where the topological SC has open boundaries along both $x$- and $y$-directions. The band structure of

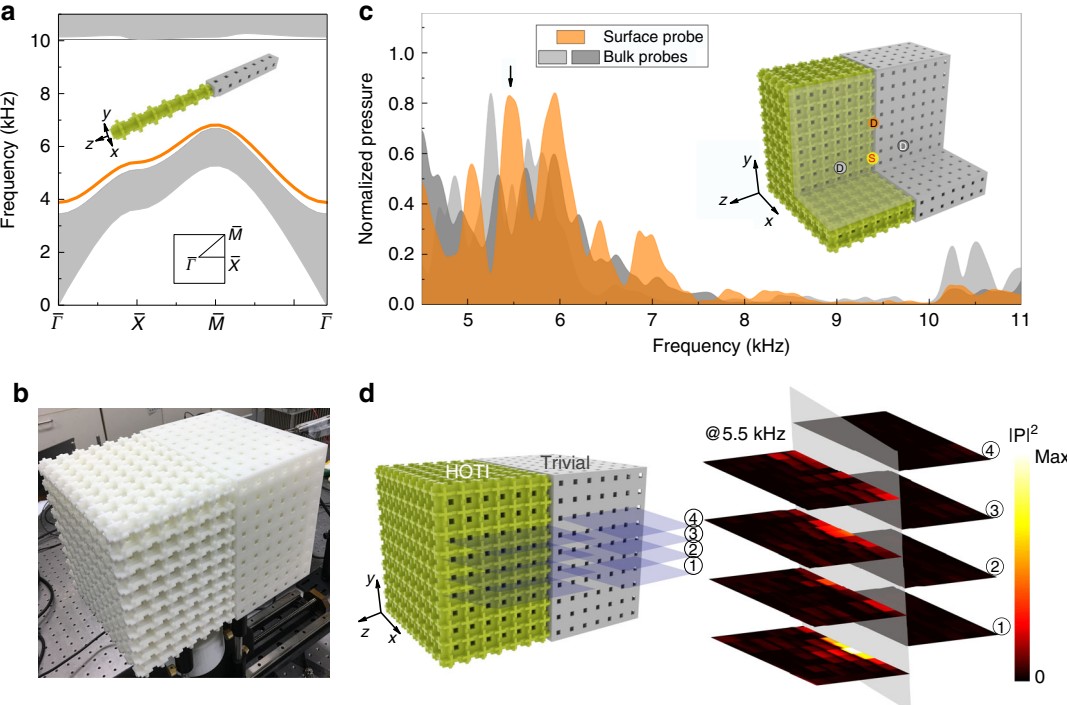

**Fig. 2** Topological surface states: boundary with codimension one. **a** Calculated band structures of the bulk (gray) and surface (orange) states for the supercell with a surface boundary in the $x$–$y$ plane between the topological and trivial SCs (see the inset). Surface Brillouin zone is shown in the inset. **b** Photograph of the fabricated surface sample, consisting of the topological and trivial SCs to form a surface boundary. **c** Measured transmission spectra for three types of pump–probe configurations denoted as the bulk and surface probes. As depicted in the inset, the source is placed on the surface. Two bulk probes are placed in the topological and trivial SC sides, respectively, while the surface probe is placed on the surface away from the source. **d** Acoustic pressure profile for the surface states measured at the excitation frequency of 5.5 kHz (indicated in **c** by the black arrow) using slice plots. The positions of the slices are depicted in the figure as well.

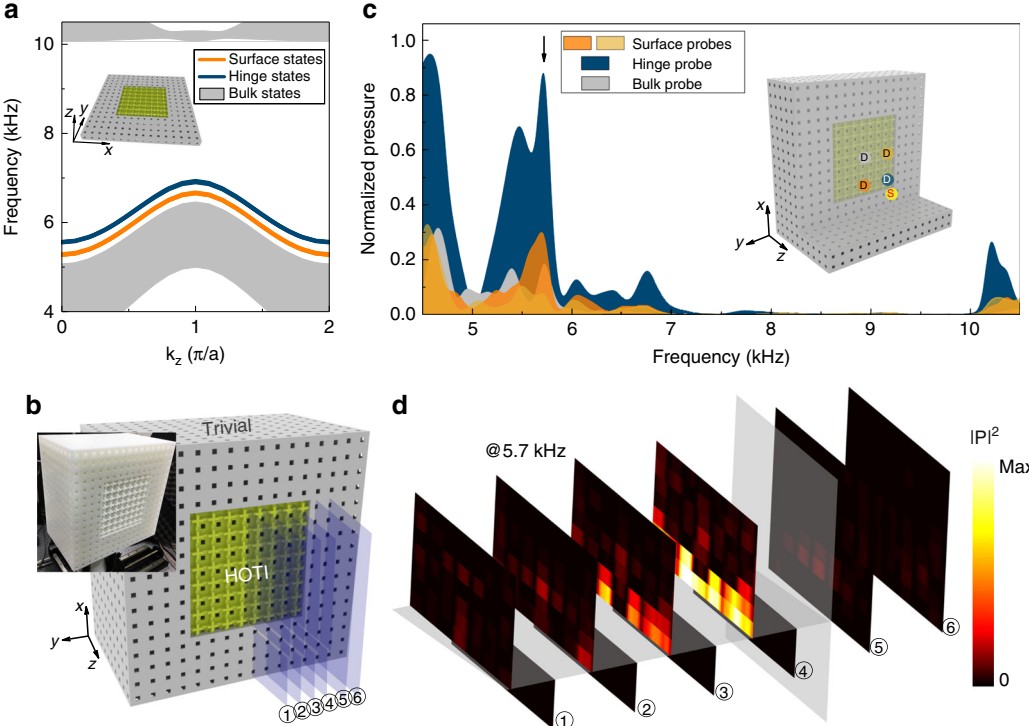

**Fig. 3 Topological hinge states: boundary with codimension two. a** Calculated acoustic dispersions of the bulk (gray), surface (orange), and hinge (dark blue) states for the supercell with both surfaces and hinges (see the inset). **b** Photograph of the fabricated hinge sample, consisting of the topological SC (with $8 \times 8 \times 10$ unit cells) surrounded by the trivial SC in both the *x*- and *y*-directions (of thickness $4a$). **c** Measured transmission spectra for four types of pump–probe configurations: the bulk probe (gray), two surface probes (orange), and the hinge probe (dark blue). The inset depicts the positions of the source and the probes for the four pump–probe configurations (the color of each probe is the same as in the transmission spectra). **d** Acoustic pressure profile for the hinge states measured at the excitation frequency of 5.7 kHz (indicated by the black arrow in **c**) and presented using slice plots (the locations of the slices are indicated in **b**).

a hinge supercell is calculated and presented in Fig. 3a. A sketch of the hinge supercell is depicted in the figure, which consists of a block of the topological SC with $8 \times 8 \times 1$ unit cells (periodic along the *z* direction) surrounded by a "wall" of the trivial SC (the thickness of the wall is $4a$) in the *x*- and *y*-directions. This supercell has four interfaces and four hinges. It is seen from the figure that there are surface states (orange curves) and hinge states (dark blue curves), as predicted by the theory. These topological surface and hinge states have four-fold degeneracy due to the coexistence of four surfaces and hinges (see Supplementary Note 5 for details).

The hinge sample fabricated in our experiment has a block of the topological SC with $8 \times 8 \times 10$ unit cells surrounded by the trivial SC (see Fig. 3b for a photograph of the sample). We perform pump–probe measurements on the hinge sample with four different configurations (see the inset of Fig. 3c). The source is placed near one of the four hinges to enhance the excitation of the hinge states. There is one detection for the bulk probe at the center of the sample, two detections near the interfaces for the surface probes, and another detection near the hinge for the hinge probe. The measured transmission spectra are presented in Fig. 3c, where the inset illustrates the source and probe locations. Several features characterize the higher-order topology in this sample. First, we observe that the frequency range of 5.3–7.1 kHz is dominated by the hinge response, where the bulk and surface responses are considerably weaker. We further measure the acoustic pressure profile of the hinge state excited at 5.7 kHz. The results are shown in Fig. 3d using the slice plots. The spatial resolution of the acoustic field scanning in the figure is 2 mm along the *x*-direction and 2 cm along the *y*- and *z*-directions. It is

observed from the figure that the acoustic energy is mostly concentrated along the hinge between the *x-z* and *y-z* interfaces, indicating the excitation of the hinge state. Away from the hinge, the acoustic energy decays quickly. To detect the surface states, we perform measurements on the excitation of the surface states in the hinge sample by putting the source to a location near the surface (see Supplementary Note 6 for the observation of surface states in the hinge sample). The coexistence of the surface and hinge states is an important feature of the higher-order topology.

We now study a "corner sample" which consists of a block of the topological SC with $8 \times 8 \times 8$ unit cells enclosed by the trivial SC in all three directions (see Supplementary Note 7 for a photograph and details of the corner sample), where the third-order topology is manifested. In order to perform measurements on the surface, hinge, and corners, we cut the corner sample and guide the acoustic waves into the sample (see Methods and Supplementary Note 7 for details). Figure 4a gives a photograph of the actual sample used in the measurements. We conduct four pump–probe measurements to detect the bulk, surface, hinge, and corner states separately (see the inset of Fig. 4b for the illustration of the positions of the source and the detection probes for the four pump–probe configurations). The source is placed not far away from the corner, hinge, and surface to ensure the excitation of the topological boundary states. The transmission spectra for the corner, hinge, surface, and bulk probes are shown in Fig. 4b. Several key-features of higher-order topology are observed. First, the appearance of a strong peak in the common spectral gap of the bulk, surface, and hinge states indicates the emergence of the corner states. Second, the concurrent emergence of the corner, hinge, and surface states in the bulk bandgap is clearly visible in

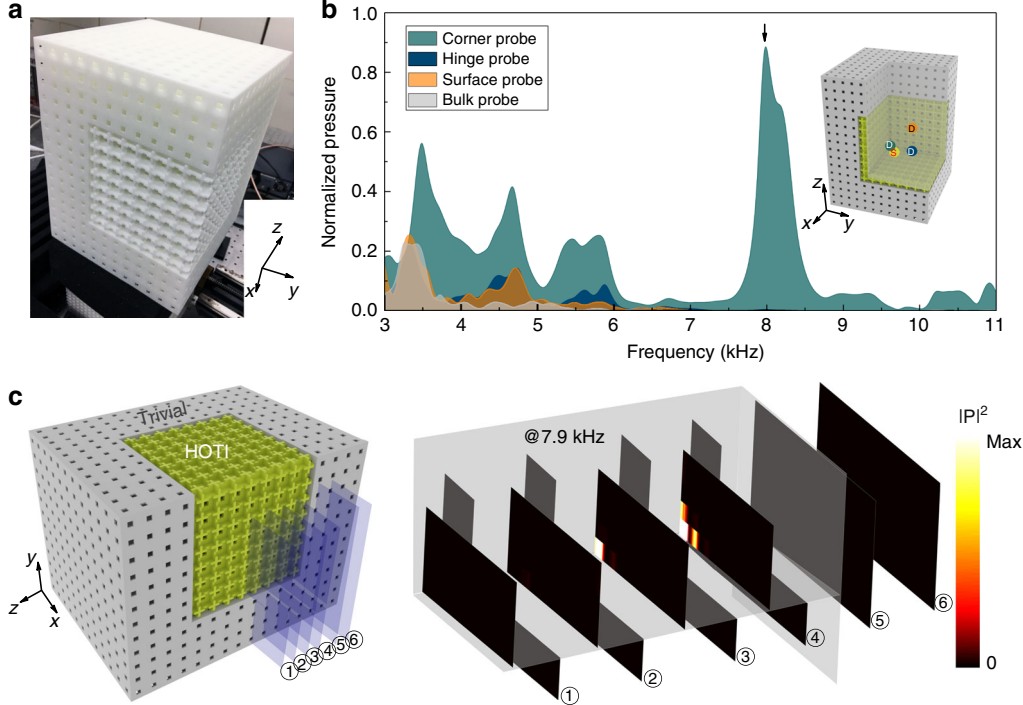

**Fig. 4** Topological corner states: boundary with codimension three. **a** Photograph of the fabricated corner sample, consisting of the topological SC (size: $8 \times 8 \times 8$ unit cells) enclosed by the trivial SC (thickness $4a$) in all three directions. To ensure pump–probe experimental measurements, the corner sample is cut to allow open interfaces, as shown in the figure. **b** Measured transmission spectra for four types of pump–probe configurations: the bulk probe (gray), the surface probe (orange), the hinge probe (dark blue), and the corner probe (dark green). The inset depicts the locations of the source and the surface, hinge, and corner probes (the same colors as in the transmission spectra). The location of the bulk probe (not shown in the inset) is in the middle of the topological SC. **c** Acoustic pressure profile for the corner states excited at the frequency of 7.9 kHz (indicated by the black arrow in **b**) and presented using slice plots (the locations of the slices are indicated in the left panel).

the transmission spectra, which is a direct evidence of the third-order topology in our 3D system. Specifically, the corner probe finds a strong peak at the frequency of 7.9 kHz, the hinge probe finds peaks ~5.6 kHz, and the surface probe finds peaks ~4.7 kHz. Furthermore, the acoustic pressure profile for the corner state excited at 7.9 kHz is presented in Fig. 4c. Typical behavior of a localized 0D mode is observed, i.e., the mode energy is concentrated around the corner and decays rapidly into the hinge, surface, and bulk. The acoustic pressure profiles for the hinge and surface states excited at their peak frequencies are shown in the Supplementary Note 8, separately, which give clear evidences of the observation of the surface and hinge states in the corner sample. In the Supplementary Note 9, we also compare the topological corner state with a normal cavity mode by numerically studying the excitation of these two types of 0D states upon multiple disorders. The results show that the topological corner state is robust against the disorders, while the cavity mode is rather sensitive to the disorders.

**Robustness**. To further characterize the symmetry-protected topological boundary states and their robustness, we introduce symmetry-preserving perturbations to the system. As emphasized above, three mirror symmetries together with the threefold rotation symmetry along the [111] direction form a full set of symmetries that protect the higher-order topology in our system. With such a set of symmetries, the only two possible gapped phases are the trivial phase with polarization $\boldsymbol{p} = (0, 0, 0)$ and the topological phase with polarization $\boldsymbol{p} = (\frac{1}{2}, \frac{1}{2}, \frac{1}{2})$. These two phases cannot be transformed into one another without breaking the symmetries listed above or closing the bandgap.

Under such symmetry constraints, we first consider inserting a layer of the topological SC or a layer of trivial SC as defect layers on the interface between the original topological SC and the trivial SC (i.e., those used in Fig. 2a) in a ribbon-like supercell. We use the arguments in ref. [35] to test the robustness of the topological surface states. Two cases are studied numerically: (i) when a layer of topological SC (with $d_{\text{intra}} = 0.875a$, depicted by the red color in Fig. 5a) is inserted, (ii) when a layer of trivial SC (with $d_{\text{intra}} = 0.125a$, depicted by the red color in Fig. 5b) is inserted. In the following, these two types of defect layers are referred as topological and trivial defect layers. We calculate the band structures of the two modified supercells, whose results are shown in Fig. 5a, b. It is seen from the figures that for both cases, the topological surface states remain in the topological band gap, although their acoustic pressure fields (i.e., the acoustic "wavefunctions") may extend into the defect layers (Fig. 5c, d). Due to the insertion of the defect layers, defect surface modes are also introduced in the high-frequency region of the bandgap (see Fig. 5a, b for their dispersions, and Fig. 5e, f for their wavefunctions). However, they do not affect the topological surface states considerably. Importantly, the topological properties of the surface states remain to be nontrivial when the defect layer is glued onto the interface. From the acoustic wavefunctions, we find that the mirror eigenvalues along the $x$- and $y$-directions $(M_x, M_x)$ for the topological surface states remain the same as those in the unperturbed structures. Specifically, the $\bar{\Gamma}$, $\bar{X}$, and $\bar{M}$ points in the surface BZ have the mirror eigenvalues of $(1, 1)$, $(-1, 1)$, and $(-1, -1)$, respectively (see Fig. 5c, d). From these mirror eigenvalues, one concludes that the topological surface states in the perturbed structures remain to have the topological polarization $\boldsymbol{p} = (\frac{1}{2}, \frac{1}{2})$, i.e., the topological surface states form an

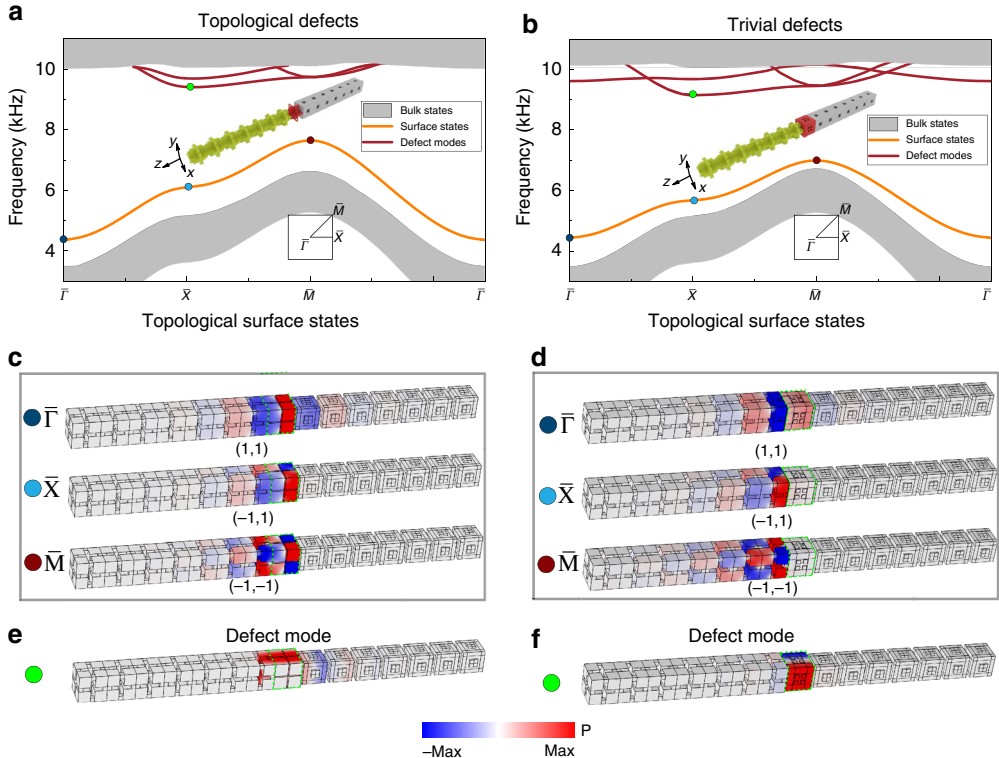

**Fig. 5** Topological surface states with the existence of symmetry-preserving defects. **a, b** The simulated band structures of a ribbon-like supercell where the symmetry-preserving defect layers with $d_{intra} = 0.875a$ and $d_{intra} = 0.125a$ are inserted, respectively. The structures of the supercells are depicted in the insets with the defect layers colored in red. The green and gray colors respectively denote the original topological ($d_{intra} = 0.85a$) and trivial ($d_{intra} = 0.15a$) SCs that have been reported in Fig. 2a. It is seen that in the bulk bandgap, the topological surface states emerge. The field distributions for these surface states at the high symmetry points of the surface Brillouin zone are presented in **c** and **d**, indicated by the colored dots (the defect layer is highlighted by the green blocks). We also label the mirror eigenvalues for the surface states in each field distribution. In addition to the topological surface states, extra defect modes are introduced (shown by the red curves in the band structures in **a** and **b**), whose field maps are presented in **e** and **f**.

effective 2D second-order topological insulator protected by the crystalline symmetry. This indicates the bulk-surface-hinge-corner correspondence is not annihilated by the symmetry-preserving perturbations. Hence, the higher-order topology is indeed protected by the set of symmetries, including the three mirror symmetries and the three-fold rotation symmetry along the [111] direction.

We also numerically study the robustness of the hinge and corner states under the symmetry-preserving perturbations. For the hinge states, band structures of the hinge supercells with the interfaces glued by the topological defect layer (left panel of Fig. 6a) and the trivial defect layer (right panel of Fig. 6a) are calculated. It is seen from the figures that the topological hinge states remain in the gap of the surface and bulk states, whose wavefunctions are localized around the hinges as shown in Fig. 6b. Here, we would like to point out that the topological hinge states still have quantized polarization along the $z$-direction, indicating that the topological hinge states form quasi-1D first-order topological insulators. This is consistent with the dimensional hierarchy of the higher-order topology. Defect modes are also introduced in the higher-frequency regime (see Fig. 6a). Their wavefuctions, however, exhibit features of surface modes rather than hinge modes due to the surface-like defect layers (see Fig. 6c).

For the corner states, due to the extremely high computational power demanded, we only compute the structures with one corner, as illustrated in Fig. 6d. Instead of the eigen-evaluations, we conduct the pump–probe simulations to detect the corner states. The calculation essentially simulates the excitation and detection of the acoustic modes near the corner, i.e., it calibrates the local density of states near the corner. An acoustic point source is placed at a location with slightly more than one lattice constant away from the "defect-corner" (represented by the corner point on the boundary between the defect layer and the trivial SC). The detection is performed at the corner (i.e., the defect corner and/or the "inner corner", with the later referring to the corner point on the boundary between the topological SC and the defect layer) and in the bulk region, respectively, for the corner and bulk probes. The calculated pump-probe transmission spectra are presented in Fig. 6e. For both cases, the corner mode is preserved.

The acoustic wavefunctions in Fig. 6f show that the corner state is still fully localized around the corner as a 0D mode in the presence of symmetry-preserving perturbations. The frequency of the corner mode in the case with the topological defect layer is 8.04 Hz, whereas it is 7.68 kHz in the case with the trivial defect layer. Both frequencies are close to the frequency of the experimentally detected corner state in the unperturbed structure, which is 7.9 kHz (see Fig. 4). These results again illustrate that the topological boundary states are robust against symmetry-preserving perturbations, consistent with the physical picture of symmetry-protected topological phases.

## Discussion

The concept of topologically robust integrated wave systems, which may inspire the next-generation technology for communication and information processing, will greatly benefit from the topological wave trapping and manipulation in multiple

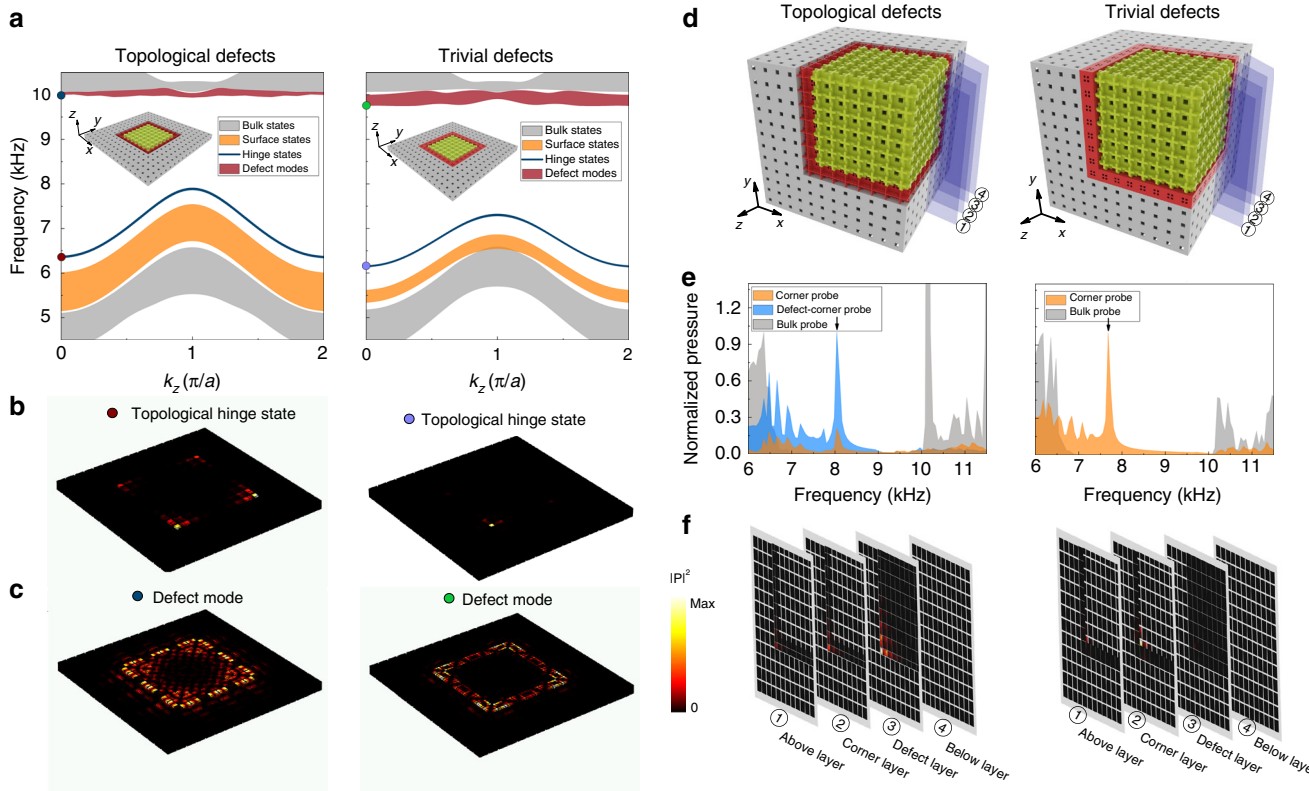

**Fig. 6** Topological hinge and corner states with the existence of symmetry-preserving defects. Two types of defects as that in Fig. 5 are studied. **a–c** The same as that presented in Fig. 5, but for the hinge supercell consisting of a topological SC with 7 × 7 unit cells, surrounded by a wall of trivial SC with four unit cells. The defect layer is inserted at the interfaces between the topological and trivial SCs (see the insets in **a** for the detailed structures). It is observed from the figures that the symmetry-preserving defects do not considerably affect the topological hinge states, which emerge and remain in the gaps of the surface and bulk states. **d** Structures with a defect layer made of topological or trivial SC and inserted on the interfaces of the corner supercell. **e** Numerically calculated transmission spectra for both the corner and bulk probes under a point source excitation (near the concerned corner). Transmission peaks corresponding to the corner modes are observed around 8.04 kHz for the case with the topological defect layer (left panel) and around 7.68 kHz for the case with the trivial defect layer (right panel). **f** Acoustic pressure profiles of the excited corner states (whose frequencies are indicated by the black arrows in **e**). The slice positions for the plots in **f** are illustrated in **d**.

dimensions such as the 2D interface channel, 1D waveguide, and 0D cavities coexisting and integrated in a single material. Here, we show that it is indeed possible to realize such a material with the help of HOTIs. Our study can be extended to photonic and elastic waves, where topologically robust integrated chips can be realized for energy-harvesting and information technology. Due to the topological nature, the fabrication challenges for frequency sensitive functional devices can be substantially reduced[44], which then opens a new route toward integrated wave systems.

*Note added*: At the final stage of this work, we notice two preprints[54,55] appeared, which realize 3D acoustic HOTIs by simulating the 3D tight-binding model in pyrochlore lattice with nearest-hopping, where different and complimentary higher-order topological phenomena are observed.

## Methods

**Experiments**. The present SCs consisting of air cavities connected by open channels are made of photosensitive resin (modulus 2765 MPa, density 1.3 g · cm$^{-3}$), which serves as acoustically hard walls. A stereo lithography apparatus is used to fabricate the samples, including a surface sample, a hinge sample, and a corner sample. The lattice constant of the SCs is $a = 2$ cm. The geometric parameters of the cavity and channel sizes are illustrated in Fig. 1b, c. The geometric tolerance is ~0.1 mm.

The experimental data on the transmission spectra in Figs. 2c, 3c, and 4b are collected using the following procedure. We eject a point-like acoustic signal, which is generated from an acoustic transducer and guided into the sample through a thin channel made of acoustically hard material (i.e., the photosensitive resin). An acoustic detector (Knowles SPM0687LR5H MEMS microphone with sizes of 4.72 mm × 3.76 mm) is used to probe the excited pressure field. Its position is

controlled by an automatic stage and can move as required. Our SCs are carefully designed and optimized (the open channel has maximal width of 6 mm) such that there is enough space for the detector to get into the sample and probe the pressure field on demand. The data were collected and analyzed using a DAQ card (NI PCI-6251). For the data in Fig. 2c, the detection positions for the bulk and surface modes are in the middle of the topological SC side, the middle of the trivial SC side, and the middle of the interface. The source for excitation of the acoustic waves is placed at a position close to the interface and nearly equal distant from the three probes. For the data in Fig. 3c, a specific hinge is considered. Given the system has $C_4$ symmetry, the study on one of the four hinges is able to provide sufficient evidence to the excitation of the hinge states. In this measurement, the source is put at a location one cell away from the center of the considered hinge and the probes for the hinge, surface, and bulk states are, respectively, performed at the center of the hinge, the center of the two interfaces intersecting at the hinge and the middle of the topological SC. For the data in Fig. 4b, similar as that for the hinge-state measurements, only one corner is considered. The source is placed two unit cells away from the corner and the locations of the corner, hinge, surface, and bulk probes are respectively at the corner, the center of the hinge along the $y$-direction, the center of the $y$-$z$ surface, and the middle of the topological SC. For the transmission spectra measurements in Supplementary Fig. 7, similar procedure is used, only the source and detecting positions are varied as required for different purposes. The said locations are all marked in the corresponding figures as insets.

For the measurement of the boundary states, the acoustic pressure distributions (Figs. 2d, 3d, and 4c, as well as those in the Supplementary Information), the above mentioned procedure is used, only the detector moves its position (controlled by the automatic stage) such that the excited modes can be spatially resolved and enough data can be collected, which are further post-processed to generate the color maps. It is worth pointing out that the straight open channels between the outer space and the inner sample space that can be accessed by the detector are separated by a lattice constant. This makes our field scanning perform with a restricted spatial step (the minimum is $a = 2$ cm) along certain directions. For example, in Fig. 2d, the scanning step along the $x$- and $y$-directions is 2 cm while

the scanning step along the z-direction can be tuned (we choose it as a fine step of 2 mm).

There are some factors that might affect our experimental measurements and cause the discrepancies between the simulations and the experimental results. First is the finite size effect. In the simulations, periodic boundary conditions are implemented while in the experiments, the samples have a finite size which can dependently shift the operating frequencies. Another factor might come from the fabrication error, which also affects the excitation of the boundary states. In addition, the physical properties of air might vary depending on the weather conditions, giving a third potential reason for the slight frequency shift between the experiments and simulations.

**Simulations**. Numerical simulations in this work are all performed using the 3D acoustic module of a commercial finite-element simulation software (COMSOL MULTIPHYSICS). The resin blocks are treated as acoustically rigid materials. The mass density and sound velocity in air are taken as $1.21\,\mathrm{kg}\cdot\mathrm{m}^{-3}$ and $343\,\mathrm{m}\cdot\mathrm{s}^{-1}$, respectively. In the eigen evaluations, all six boundaries of the unit cells are set as Floquet periodic boundaries for the data in Fig. 1b–e, as well as in the Supplementary Fig. 1. The boundaries of the supercells are set as Floquet periodic boundaries along the boundary directions, with the perpendicular directions set as plane wave radiation boundaries, for the data in Figs. 2a and 3a. The eigen evaluations in Supplementary Figs. 3, 5, and 6 also obey similar set-up. In the simulations on the excitations, all boundaries are set as plane wave radiation boundaries.

## Data availability
The data that support the plots within this work and other related findings are available from the corresponding authors upon reasonable request.

## Code availability
Numerical simulations in this work are all performed using the 3D acoustic module of a commercial finite-element simulation software (COMSOL MULTIPHYSICS). All related codes can be built with the instructions in the Method section.

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

## Acknowledgements

X.J.Z., B.Y.X., H.F.W., X.Y.X., Y.T., M.-H.L., and Y.-F.C. are supported by the National Natural Science Foundation of China (grant nos. 51902151, 11625418, 11890700, 51732006, and 51902151), the National Key R&D Program of China (2017YFA0303702, 2018YFA0306200), and the Natural Science Foundation of Jiangsu Province (grant no. BK20190284). J.-H.J. is supported by the National Natural Science Foundation of China (grant no. 11675116), the Jiangsu distinguished professor funding and a project funded by the Priority Academic Program Development of Jiangsu Higher Education Institutions (PAPD). X.J.Z. thanks H. Ge and S. Yu for their support and assistance with the experimental measurements. B.Y.X. thanks Yuxin Zhao for helpful discussions.

## Author contributions

X.J.Z., B.Y.X., and H.F.W. conceived the idea. B.Y.X. and H.F.W. did the theoretical analyses. X.J.Z. performed the numerical simulations. X.J.Z., X.Y.X., and Y.T. performed experimental measurements. J.-H.J., M.-H.L., and Y.-F.C. guided the research. All authors contributed to discussions of the results and paper preparation. J.-H.J., B.Y.X., X.J.Z., and M.-H.L. wrote the paper.

## Competing interests

The authors declare no competing interests.
