## [Peer Review File · Nature Communications]

Reviewers' Comments:

Reviewer #1:

Remarks to the Author:

The authors present a design of a higher-order acoustic topological insulator, which carries at the same time surface states, hinge states and corner states. They made the acoustic crystal according to their design and constructed different samples in which various boundary modes can be detected and characterized. They did measurements and successfully verified their predictions. Their system is a simple cubic structure, with three independent mirror symmetries (M_x , M_y and M_z). Different from the mechanisms proposed in Ref. 41 (Phys. Rev. B 96, 245115, where electric dipole, quadrupole and octupole give rise to the nontrivial surface, hinge and corner states), the mechanism here is comparably more straightforward, but it does carry all the interesting signatures of a higher-order acoustic topological insulator. The real achievement is the experimental demonstration. The theory part is also fairly complete, including the discussion of using mirror symmetries to protect the topological phases and the calculation of 3D Zak phases, which serve as the bulk topological characteristics. I believe that paper reaches the expectation of Nature Communication, and I recommend its acceptance.

Here are some minor issues for the authors' consideration:

- Why should the structures shown Fig. 1a and Fig. 1b have exactly the same bulk band structure? To me, this is possible if the 1a and 1b structures are actually the same structure, just the choice of unit cells is different. But the structures shown in the insets do not appear to be the same, at least not to my eyes. If they are not the same structure, how can the band dispersion be exactly the same? I actually find the unit cell inset in Fig. 1a quite hard to follow. Readers will have difficulty visualizing the detailed structure inside the unit cell, such as the shape and position of the air cavities and channels. The authors may want to find a better way to help readers understand the structures.
- For Figs. 1a-b, showing the mirror parities of eigen-states at different high-symmetric points, the authors should specify the band number. I suppose that they are working with the lowest band. The same query for the Wannier band calculations.
- At the first glance, I thought that the mechanism is based essentially on a 3D SSH model, but the authors emphasized in the later part of the paper and in the appendix that it is NOT a 3D SSH. Can the authors explain in simple terms what then is the mechanism (may be in terms of a tight-binding model)?
- The authors mention Wyckoff positions several times. It would be good if they include a figure that indicate the Wyckoff positions. And as the higher-order topological order is protected by crystalline symmetries here, it would be nice if the authors can show the space group number of their structure.
- The authors may want to give some discussions to distinguish between the topological corner states and normal (trivial) cavity modes. This can help general readers appreciate the importance of their work.
- The manuscript can use some English editing.

Reviewer #2:

Remarks to the Author:

I was asked to specifically comment on whether the model studied in the manuscript indeed realizes a higher-order TI. To me, this is currently not demonstrated convincingly in the manuscript and the SM, but I see a chance that the authors can add this information in a revision.

Specifically, I take as a definition of a HOTI that there exists a bulk topological invariant that allows to predict the existence of, for instance, corner modes in a geometry where the defining symmetries of the phase are preserved in open boundary conditions. This bulk-boundary correspondence should be robust against any symmetry-preserving boundary deformations. Such protection necessarily involves a non-local (spatial) symmetries.

Taking the example of a 3rd order 3D HOTI discussed in the manuscript, the mirror symmetries referred to by the authors do not protect the corner modes. One can "glue" a 2nd order 2D HOTI to two opposite of the surfaces and this way remove all the 8 corner modes by hybridization while preserving all mirror symmetries. The authors should come up with a symmetry that prevents such a process from being possible and thus truly protects the corner modes.

We thank all the reviewers for their valuable comments and suggestions. The paper is revised considerably according to those comments and suggestions. The revisions are marked in blue in the revised manuscript and the supplementary information.

Reply to the Reviewer #1

Reviewer's remarks and comments: *“The authors present a design of a higher-order acoustic topological insulator, which carries at the same time surface states, hinge states and corner states. They made the acoustic crystal according to their design and constructed different samples in which various boundary modes can be detected and characterized. They did measurements and successfully verified their predictions. Their system is a simple cubic structure, with three independent mirror symmetries (M_x , M_y and M_z). Different from the mechanisms proposed in Ref. 41 (Phys. Rev. B 96, 245115, where electric dipole, quadrupole and octupole give rise to the nontrivial surface, hinge and corner states), the mechanism here is comparably more straightforward, but it does carry all the interesting signatures of a higher-order acoustic topological insulator. The real achievement is the experimental demonstration. The theory part is also fairly complete, including the discussion of using mirror symmetries to protect the topological phases and the calculation of 3D Zak phases, which serve as the bulk topological characteristics. I believe that paper reaches the expectation of Nature Communication, and I recommend its acceptance. Here are some minor issues for the authors' consideration: • Why should the structures shown Fig. 1a and Fig. 1b have exactly the same bulk band structure? To me, this is possible if the 1a and 1b structures are actually the same structure, just the choice of unit cells is different. But the structures shown in the insets do not appear to be the same, at least not to my eyes. If they are not the same structure, how can the band dispersion be exactly the same? I actually find the unit cell inset in Fig. 1a quite hard to follow. Readers will have difficulty visualizing the detailed structure inside the unit cell, such as the shape and position of the air cavities and channels. The authors may want to find a better way to help readers understand the structures. • For Figs. 1a-b, showing the mirror parities of eigen-states at different high-symmetric points, the authors should specify the band number. I suppose that they are working with the lowest band. The same query for the Wannier band calculations.”*

Our reply: We thank the reviewer for these useful comments and suggestions. We agree that the first version of Fig. 1a-1b is not clear in the geometry, in particular how the trivial and topological sonic crystals are geometrically related to each other. Based on the reviewer's comments, we have revised Fig. 1 carefully in the new manuscript. Specifically, we added the motherboard structure to the figure where the air cavities have a center-to-center distance of $d = 0.5a$. There are two kinds of center-to-center distances: the first one involves the links within the unit cell, denoted as d_{intra} ; the other one involves the links between the adjacent unit cells, denoted as d_{inter} ; these two distances are related to each other by $d_{intra} + d_{inter} = a$. For

the motherboard sonic crystal, $d_{intra} = d_{inter} = 0.5a$. By reducing the center-to-center distance within the unit cell to $d_{intra} < 0.5a$ ($0.15a$ in our case), we reach at the lattice with trivial topology where the Wannier center is located at the unit cell center. In contrast, by increasing the center-to-center distance within the unit cell to $d_{intra} > 0.5a$ ($0.85a$ in our case), we arrive at the lattice with nontrivial topology where the Wannier center is located at the corner of the unit cell. We denote those two types of sonic crystals respectively as the shrunken and expanded sonic crystals in the main text. Because their structures differ only by a translation of the unit cell center, the acoustic band structures for those two sonic crystals are identical, yet the band topology is different. This situation is similar to the trivial and topological bands in the SSH model.

The above scenario is revealed through the revised Fig. 1 where the distance d_{intra} is depicted clearly and the evolution from the motherboard sonic crystal with $d_{intra} = 0.5a$ to the shrunken and expanded lattices is also illustrated clearly. We also added several insets to depict the structure of the motherboard, the trivial and the topological sonic crystals, which will make the 3D figures more understandable. The main text and the caption are revised accordingly to ensure the above physical picture is clearly presented. In addition, we emphasized in the main text and the caption that we study only the lowest acoustic band which is the only band below the first acoustic band gap.

Reviewer's comments: “• *At the first glance, I thought that the mechanism is based essentially on a 3D SSH model, but the authors emphasized in the later part of the paper and in the appendix that it is NOT a 3D SSH. Can the authors explain in simple terms what then is the mechanism (may be in terms of a tight-binding model)?*”

Our reply: We thank the reviewer for these comments. The underlying physics is indeed the same as the 3D SSH model, which is essentially characterized by the 3D Zak phase or the dipole polarization associated with it. However, there are some crucial differences between our system and the 3D tight-binding SSH model, as revealed in details in both the original and revised manuscript (as well as the Supplementary Information). For instance, the acoustic bands break the chiral (or sublattice) symmetry and evolve into a linear dispersion when the frequency goes to zero. The underlying microscopic mechanism is that the couplings beyond the nearest-neighbors exist in our acoustic system. A 3D tight-binding SSH model with only nearest-neighbor coupling, where the chiral symmetry is preserved, cannot reproduce such linear dispersions. This leads to a more crucial difference for structures that are finite in all three directions. For the 3D tight-binding model, the corner states are spectrally buried in the surface and hinge states. As a result, the corner states are unable to be distinguished from the surface and hinge states in the energy spectra. In contrast, for our acoustic system, the corner, hinge and surface states are spectrally separated from each other, making them much easier to be accessed independently. Such a spectral separation of the topological surface, hinge and corner states is important for the experimental demonstration of the higher-order topology.

Reviewer's comments: “• *The authors mention Wyckoff positions several times. It would be good if they include a figure that indicate the Wyckoff positions. And as the higher-order topological order is protected by crystalline symmetries here, it would be nice if the authors can show the space group number of their structure.*

• *The authors may want to give some discussions to distinguish between the topological corner states and normal (trivial) cavity modes. This can help general readers appreciate the importance of their work.*

• *The manuscript can use some English editing.”*

Our reply: We thank the reviewer for these useful suggestions. We have included an extra figure that indicates the maximal Wyckoff positions in the revised Supplementary Information. Our designed sonic crystals have simple cubic geometry, belonging to the space group P_{m3m} (no. 221), which has been highlighted in the revised manuscript. To address the difference between the topological corner states in our system and the normal cavity modes, we create a cavity in the trivial sonic crystal. The details are elaborated in the revised Supplementary Information. We compare the robustness of the topological corner state and the cavity mode through numerical simulations. Specifically, we consider the geometry deformation at/near the corner or the cavity (see the revised Supplementary Information). We find that the topological corner state has a rather small frequency shift (<3%) upon various distortions while the cavity mode is quite sensitive to the same distortions (with frequency shift ~10% or >10%). These results indicate that the topological corner state is indeed more robust than the normal cavity mode, which may inspire new design strategy for robust localized modes protected by higher-order topology. Finally, we carefully and thoroughly revised the manuscript and Supplementary Information to improve the presentation quality.

Reply to the Reviewer #2

Reviewer's remarks and comments: *"I was asked to specifically comment on whether the model studied in the manuscript indeed realizes a higher-order TI. To me, this is currently not demonstrated convincingly in the manuscript and the SM, but I see a chance that the authors can add this information in a revision. Specifically, I take as a definition of a HOTI that there exists a bulk topological invariant that allows to predict the existence of, for instance, corner modes in a geometry where the defining symmetries of the phase are preserved in open boundary conditions. This bulk-boundary correspondence should be robust against any symmetry-preserving boundary deformations. Such protection necessarily involves a non-local (spatial) symmetries. Taking the example of a 3rd order 3D HOTI discussed in the manuscript, the mirror symmetries referred to by the authors do not protect the corner modes. One can "glue" a 2nd order 2D HOTI to two opposite of the surfaces and this way remove all the 8 corner modes by hybridization while preserving all mirror symmetries. The authors should come up with a symmetry that prevents such a process from being possible and thus truly protects the corner modes."*

Our reply: We thank the reviewer for these valuable comments. The topology of our system is essentially the same as the 3D SSH model. Based on the modern polarization theory, the Zak phase for such a system is quantized by the mirror symmetries, which can fully determine the field polarization and hence the bulk-boundary correspondence (Refs. 3 and 4 in the Supplementary Information). In our topological sonic crystal, the Zak phase is calculated as (π, π, π) , assuming the origin point is at the center of the unit cell. This indicates non-zero bulk polarization (0.5, 0.5, 0.5), which pins the Wannier center (WC) at the corner of the unit cell. The local displacement of a WC from the origin point represents a microscopic map of the local polarization field. Then the sum of these local polarization fields leads to the macroscopic quantum mechanical polarization of the system. Correspondingly, the WC in our topological sonic crystal leads to polarization fields localized on the surface, on the hinge or at the corner, respectively as the manifestation of the first-, second- or third-order topology (see Fig. 1f or the Supplementary Information for more details). Indeed, the topological surface, hinge and corner states have been observed in our experiments. From the perspective of the surface impedance theory, it has also been proved rigorously that the relation between the bulk bands and the topological boundary states can be explained by the Zak phase, which is protected by the inversion symmetry (Ref. 53). If one applies consecutively the argument of Ref. 53, topological surface, hinge and corner states will emerge (as illustrated in Fig. 1f). Similar arguments have been also applied in Ref. 31, a pioneering work in the field of higher-order topological insulators.

The idea of testing the topology by continuous deformations that respect the crystalline symmetry is indeed crucial for the definition of topological crystalline insulators. In the manuscript, we have shown that by tuning the geometric parameters, the sonic crystal (which always respects the mirrors symmetries) is deformed from the topological phase into the trivial phase, associated with a bandgap-closing. This agrees with the standard picture of topological

crystalline insulators where the band topology is protected by the crystalline symmetry. However, the test of gluing a 2nd order TI to the surface of the 3rd order TI (as depicted by Fig. 1Ra) is normally not considered as the test for the stability of the higher-order topological insulators. This is because such surface deformations are not compatible with the definition of symmetry-preserving perturbations.

We can imagine several other cases where such surface deformations can destroy the topological surface states: (i) For weak topological insulators that have two Dirac cones on the surface, gluing the surface with a 2D system with two Dirac cones may gap the topological surface Dirac cones. (ii) For the SSH model in 1D, one can easily annihilate the end states by gluing a 0D perturbation to each end. As shown in Fig. 1Rb-c, after the gluing, the topologically nontrivial 1D SSH chain (Fig. 1Rb) becomes a topologically trivial chain (Fig. 1Rc), where the original end states disappear and the system supports only the bulk states. The case (ii) is a particularly close analog of the scenario happening in our 3rd order TI, revealing the contradiction between the well-defined bulk topology protected by inversion symmetry and the surface-gluing gedanken experiments. This is because the surface-gluing procedure introduces deformation that is incompatible with the crystalline symmetry (e.g., the inversion symmetry), which is defined as the symmetry of the unit cell (denoted here as ‘local’) rather than that of the finite (supercell) structures (denoted here as ‘global’). With such an understanding, the gedanken experiment of surface-gluing in fact contains perturbations that break the crystalline symmetry. This naturally explains why surface-gluing procedures can annihilate the topological end states in the 1D SSH model without changing the bulk bands or breaking the global symmetry (but does break the local symmetry). Similarly, the perturbation of gluing a 2nd order 2D HOTI to two opposite surfaces of the 3rd order 3D HOTI will naturally break the local inversion symmetry of the unit cells at the two open boundaries in the direction perpendicular to the 2D HOTI plane (see Fig.1Ra). As a result, the bulk-boundary correspondence may break down, as the perturbations essentially break the crystalline symmetry. Such a scenario, however, does not rule out the symmetry-protected topology in the normal and higher-order topological crystalline insulators. Nevertheless, in light of the reviewer’s comments, we have emphasized in the revised manuscript that the mirror symmetries essentially protect the higher-order topology.

Figure R1 | The surface-gluing gedanken experiment and its effect. **a**, Schematic of the gedanken experiment, where two 2nd order 2D HOTIs are glued to the two opposite surfaces of the 3D HOTI. A projection along the perpendicular (to the 2D HOTI plane) direction is illustrated using a 1D SSH chain, where the boundary unit cells and their mirror planes are indicated. The introduction of the two 2D HOTIs essentially creates perturbations that break the local mirror symmetry. **b**, A 1D topological SSH chain without perturbations and the field maps of its two end states. t_b and t_b respectively illustrate the intra and inter site-to-site distances. The colored regions in the field maps represent the magnitude of the field intensity along the 1D chain: the larger the region is, the higher the intensity is. **c**, A trivial 1D SSH chain created by gluing a 0D perturbation with $t = t_b$ to each end. Such perturbations break the local mirror symmetry and hence the bulk-boundary correspondence. As a result, only bulk states survive in this case. Note that by introducing the perturbations, four bulk states emerge, while in **b** only two end states existed.

Reviewers' Comments:

Reviewer #1:

Remarks to the Author:

The authors have addressed all my comments and suggestions and the revised manuscript is in good shape. I recommend acceptance.

Reviewer #2:

Remarks to the Author:

Core point of my criticism and the author's response is the question whether higher-order topology, or topological phases in general, should be robust against 'glueing' symmetry-respecting systems to the boundary. I argue that this indeed a stability criterion that topological phases of matter should satisfy.

For higher-order topology, concretely, this is detailed in Ref. [35] in which the terminology of higher-order topology was introduced. For instance, the stability of a phase with chiral hinge modes is argued as follows: "The minimal relevant surface perturbation of that kind is the addition of an integer quantum Hall (or Chern insulator) layer ..."

The authors give two examples which are meant to demonstrate that the glueing procedure is too strong of a constraint. Both of these examples are not valid, as I will argue below. The idea behind the 'boundary glueing' arguments is that symmetry-respecting boundary disorder should not destroy the topology. But of course it could 'passivate' the surface by introducing a dead layer. The formation of such a dead layer would then move the topological boundary modes towards the bulk of the sample, but it would not annihilate them.

(i) weak topological insulator: The weak topology is protected by time-reversal AND translation symmetry. The authors propose to add a two-dimensional system with two Dirac cones to the surface of a weak TI to gap out its surface Dirac cones. In fact, such a two-dimensional system does not exist, when spinful time-reversal and translational symmetry are to be respected. (If the authors are of a different opinion, I ask them to name such a phase.) Importantly, the two topological Dirac cones of weak TI surface appear at two distinct time-reversal symmetric momenta in the surface Brillouin zone.

(ii) SSH model: The topology of the SSH model is protected by chiral symmetry. When degrees of freedom are added to the end of the chain, these must be a singlet under chiral symmetry (i.e., the trace of the chiral symmetry over the added degrees of freedom glued to one end must be 0). The example given in the authors' reply does not meet this criterion.

In view of these points, I do not think that the author's reply adequately addresses my concern.

We thank all the reviewers for their valuable comments and suggestions. The paper is revised according to those comments and suggestions. The revisions are marked in blue in the revised manuscript and the supplementary information.

Reply to the Reviewer #1

Reviewer's remarks and comments: *"The authors have addressed all my comments and suggestions and the revised manuscript is in good shape. I recommend acceptance."*

Our reply: We thank the reviewer for the valuable time and efforts put in reviewing our manuscript. The helpful comments have enabled us to further boost the scientific merits of our manuscript.

Reply to Reviewer #2

Reviewer's comments: *Core point of my criticism and the author's response is the question whether higher-order topology, or topological phases in general, should be robust against 'glueing' symmetry-respecting systems to the boundary. I argue that this indeed a stability criterion that topological phases of matter should satisfy. For higher-order topology, concretely, this is detailed in Ref. [35] in which the terminology of higher-order topology was introduced. For instance, the stability of a phase with chiral hinge modes is argued as follows: "The minimal relevant surface perturbation of that kind is the addition of an integer quantum Hall (or Chern insulator) layer ..." The authors give two examples which are meant to demonstrate that the glueing procedure is too strong of a constraint. Both of these examples are not valid, as I will argue below. The idea behind the 'boundary glueing' arguments is that symmetry-respecting boundary disorder should not destroy the topology. But of course it could 'passivate' the surface by introducing a dead layer. The formation of such a dead layer would then move the topological boundary modes towards the bulk of the sample, but it would not annihilate them.... In view of these points, I do not think that the author's reply adequately addresses my concern.*

Our reply: We thank the reviewer for explaining his/her comments and remarks in more details. After carefully reading these comments and remarks, we realize that we misunderstood the reviewer's ideas. What we worried was to attach "fractionalized" systems to the surface which will certainly annihilate the surface states and then the corner states. For instance, if only one atom (i.e., half of the unit cell) is added to each boundary of a topological SSH chain, the end states will be removed. The two examples we gave in the last reply reflect our worries on gluing "fractionalized" systems to the surface, which are the consequences of our misunderstanding of the reviewer's points. As clarified by the reviewer in the present report, the perturbations glued on the surfaces must not be such fractionalized systems, but fulfill all the

symmetries required to protect the topological phase, which we now understand as the standard definition of higher-order topological insulators (as elaborated in Ref. [35]). We also agree that the three mirror symmetries are not enough to protect the surface states and it is crucially important to point out the full set of symmetries that protect the higher-order topology in our system.

In the revised manuscript, we state clearly that the three mirror symmetries together with the three-fold rotation symmetry along the [111] direction form a full set of symmetries that protect the higher-order topology in our system. With such a set of symmetries, the only two possible gapped phases are the trivial phase with polarization $\vec{p} = (0,0,0)$ and the topological phase with polarization $\vec{p} = (\frac{1}{2}, \frac{1}{2}, \frac{1}{2})$. These two phases cannot be transformed into one another without breaking the symmetries listed above or closing the band gap. The symmetry constraints reduce the number of possible gapped phases and complete the picture of the symmetry-protected topological phase. In this way, the only allowed symmetry-respecting perturbations in this system are the trivial and topological sonic crystals.

We then consider attaching a layer of the trivial sonic crystal or a layer of topological sonic crystal on all the six surfaces. Due to the extremely high computational power demanded, we can only compute the structure with one corner, which is illustrated in Fig. A1 in this reply. We study two cases numerically: (i) when a layer of topological sonic crystal (with $d_{intra} = 0.875a$, depicted by the red color in Fig. A1a) is glued to the interfaces between the original trivial ($d_{intra} = 0.15a$) and topological ($d_{intra} = 0.85a$) sonic crystals, (ii) when a layer of trivial sonic crystal (with $d_{intra} = 0.125a$, depicted by the red color in Fig. A1d) is glued to the interfaces between the original trivial and topological sonic crystals.

We use the pump-probe simulation to detect the corner states. The calculation essentially simulates the excitation and detection of the acoustic modes near the corner, i.e., it calibrates the local density of states near the corner. An acoustic point source is placed at a location with slightly more than one lattice constant away from the “defect-corner” (i.e., the corner point on the boundary between the defect layers and the trivial sonic crystal). To optimize the pump-probe study for the corner states, in Fig. A1a, the detections are performed at the defect corner and in the bulk region respectively for the corner and bulk probes, while in Fig. A1d, the corner detection is performed at the corner point on the boundary between the topological sonic crystal and the defect layers. The calculated pump-probe transmission spectra are presented in Figs. A1b and A1e for the cases of the topological defect layer and the trivial defect layer, respectively. For both cases (i) and (ii), the corner mode is preserved when the symmetry-respecting perturbations are introduced.

The acoustic pressure profiles (i.e., the acoustic “wavefunctions”) in Figs. A1c and A1f show that for both cases (i) and (ii), the corner state is still fully localized around the corner as a 0D mode. The wavefunctions of the corner state for the two cases are different: for the case with trivial defect layer, the corner state is mainly localized in the topological sonic crystal; whereas for the case with topological defect layer, the corner state extends into the defect layer considerably. Despite such a change in the wavefunction, the corner mode remains to be a 0D localized state within the topological band gap. In addition, the frequency of the corner mode in case (i) is 8.04 kHz, while in case (ii) it is 7.68 kHz. Both frequencies are very close to the frequency of the experimentally detected corner frequency 7.9 kHz in the unperturbed structure (see Fig. 4 of the main text). These results indicate that the corner state is quite robust against symmetry-preserving perturbations and thus verify the symmetry-protected topology from realistic simulations.

Figure A1| Topological corner states with the existence of the symmetry-respecting defects.

a, A layer of topological defect respecting all four symmetries is introduced to the unperturbed corner structure. The defect layer is sketched in red color, with $d_{intra} = 0.875a$. The green and grey colors respectively denote the original topological ($d_{intra} = 0.85a$) and trivial ($d_{intra} = 0.15a$) sonic crystals that have been reported in the main text. **b**, Numerically calculated transmission spectra for both the corner and bulk probes under a point source excitation (near the concerned corner). A transmission peak is observed around 8.04 kHz, where the corner state is excited in the bulk band gap. Note in **a**, the corner probe is conducted at the defect corner due to the fact that the corner state extends to the defect layer, as shown in **c** for the excited corner mode (indicated by the black arrow in **b**). The slice positions for the plot in **c** are illustrated in **a**. **d-f**, The same as **a-c**, only for the case with trivial defect ($d_{intra} = 0.125a$). It is seen that similar as the topological defect, the trivial defect does not destroy the corner mode either (its frequency is 7.68 kHz).

We also numerically study the robustness of the surface states under the symmetry-preserving perturbations. We calculate the band structures of the ribbon-like supercells composed of the topological sonic crystal and the trivial sonic crystal, with the defect layer inserted at the interface between them. The results are shown in Fig. A2. The studied supercells are illustrated in the insets of Figs. A2a and A2d. It is seen from the figures that for both cases, the topological surface states remain in the topological band gap (Figs. A2a and A2d), although the acoustic wavefunctions may extend into the defect layer (Figs. A2b and A2e). Defect surface modes are also introduced close to the high-frequency part of the band gap (see Figs. A2a and A2d for their dispersions and Figs. A2c and A2f for their wavefunctions). However, they do not considerably affect the topological surface states. Importantly, the topological properties of the topological surface states remain to be nontrivial when the defect layer is glued on the interfaces. From the acoustic wavefunctions, we find that the mirror eigenvalues along the x and y directions (M_x, M_x) for the topological surface states remain the same as those in the unperturbed structure. Specifically, the $\bar{\Gamma}$, \bar{X} , and \bar{M} points in the surface Brillouin zone have the mirror eigenvalues of $(1, 1)$, $(-1, 1)$ and $(-1, -1)$, respectively (see Figs. A2b and A2e). From these mirror eigenvalues, one concludes that the topological surface states in the perturbed structures remain to have the topological polarization $(\frac{1}{2}, \frac{1}{2})$, i.e., the topological surface states form an effective 2D second-order topological insulator protected by the crystalline symmetry. This again indicates the bulk-surface-hinge-corner correspondence is not annihilated by the symmetry-preserving perturbations. Hence, the higher-order topology is indeed protected by the set of symmetries including the three mirror symmetries and the

three-fold rotation symmetry along the [111] direction.

The above results can also be understood from the aspect of adiabatic tuning: a defect layer made of the trivial sonic crystal can be adiabatically tuned into the trivial sonic crystal in the original structure without breaking the symmetry or closing the band gap. In this process, the topological surface states remain in the band gap, while the defect surface states are tuned into the bulk bands. Similarly, a defect layer made of the topological sonic crystal can be adiabatically tuned into the topological sonic crystal in the original structure without breaking the symmetry or closing the band gap. In the process, the topological surface states remain in the band gap, while the defect surface states are tuned into the bulk bands. Following these arguments, one can also prove the robustness of the topological hinge states and corner states that are protected by the symmetries and the band gap.

Revisions: In the revised manuscript, we have added in the main text the following sentence in the theory part: “The higher-order band topology is protected by a set of crystalline symmetries including the three mirror symmetries and the C_3 rotation symmetry along the [111] direction.”. We have also added the results presented in this reply to the revised Supplementary Information.

Figure A2| Topological surface states with the existence of symmetry-respecting defects. Two types of symmetry-respecting defects as in Fig. A1 are introduced to the interface between the original topological and trivial sonic crystals. The same color notations as that in Fig. A1 are used.

a and d, The simulated projected band structures for the ribbon-like supercells with the topological and trivial defects, respectively. It is seen that in the bulk band gap, the topological surface states emerge (with their frequencies slightly deviated from the original surface states in Fig. 2a of the main text). The field distributions for these surface states at different high symmetry points are presented in **b** and **e**, indicated by the colored dots (the defect layers are highlighted by green blocks). We also label the mirror eigenvalues for the surface states in each field map. In addition to the topological surface states, extra defect modes are introduced (shown by the red curves in **a** and **d**). Different from the topological surface states, the defect modes are localized in the defect layer, regardless of whether topological or trivial defects are induced (see the field maps for the defect modes in **c** and **f**).

Reviewers' Comments:

Reviewer #2:

Remarks to the Author:

I appreciate the detailed response of the authors to my second report. This eliminates the misunderstanding we had in the first round and correctly addresses the question about symmetry.

I have one small final remark regarding Fig 3a: I do not understand why the surface and hing states both have a dispersion similar to a 1D mode. Should not the surface state be dispersive along both inplane momenta of the surface and hence occupy a finite width in frequency (as the bulk bands to) rather than just being represented by a single line?

Aside from this point I consider the manuscript ready for publication now.

We thank the reviewer for the valuable comments and suggestions in the previous and present review rounds, which have helped us considerably improve the quality of this work. Based on the comments in the present round, as well as the editorial requests, the manuscript and the Supplementary Information are revised accordingly. The revisions are marked in blue.

Reply to Reviewer #2

Reviewer's comments: *I appreciate the detailed response of the authors to my second report. This eliminates the misunderstanding we had in the first round and correctly addresses the question about symmetry. I have one small final remark regarding Fig 3a: I do not understand why the surface and hinge states both have a dispersion similar to a 1D mode. Should not the surface state be dispersive along both inplane momenta of the surface and hence occupy a finite width in frequency (as the bulk bands do) rather than just being represented by a single line? Aside from this point I consider the manuscript ready for publication now.*

Our reply: We thank the reviewer for being meticulous and patient. The detailed explanation of his/her comments and remarks in the previous round was very clear, inspiring and helpful, which we appreciate very much.

Regarding the surface states in Fig. 3a, they are actually four degenerate states for each k_z , as presented in the Supplementary Information (see Supplementary Note 5 and Supplementary Figure 5). As suggested by the reviewer, the surface states indeed should be dispersive along both k_x and k_y directions of the surfaces. However, due to the limited computational power, the topological sonic crystal in our hinge sample consists of 8×8 unit cells (surrounded by a layer of trivial sonic crystal with 4 unit cells). Such a small size scale can only allow the existence of the fundamental surface states (i.e., the surface states with the smallest values of possible k_x and k_y) and hence there are only four degenerate surface states observed in our system, consistent with the four interfaces constructed between the topological and trivial sonic crystals. If we increase the sample size (e.g., the number of unit cells in the topological sonic crystal becomes 12×12), it is found that the surface states with higher- k begin to emerge and they may also interact with the lower- k components of the surface states. As a result, the surface states occupy a finite width in frequency (see Fig. A1 below), just as pointed out by the reviewer. Meanwhile, the hinge states remain to be four degenerate states localized on the hinges, as shown by the pressure field maps presented in Fig. A1. To provide a clearer picture, we have included the study of the large hinge sample (i.e., the 12×12 unit-cell sample) into the Supplementary Information.

Revisions: In the revised Supplementary Information, we have included the study of the hinge sample with 12×12 unit cells in Supplementary Note 5. An extra figure (Fig. A1 or the Supplementary Figure 6) is also added.

Figure A1| Topological hinge and surface states in the hinge supercell with 12×12 unit cells. The calculated eigen spectra of the hinge supercell are presented. The simulation set-up is the same as that in Fig. 3a of the main text. It is seen that in this sample, the surface states occupy a finite frequency range, different from the case in the smaller sample (i.e., the 8×8 unit-cell sample presented in Fig. 3a of the main text) where only four degenerate surface states are found. This is because the present sample supports both fundamental and higher- k surface states, which are split in frequency, as shown by the surface state field maps (represented by the light and dark orange dots). One surface state locates at 5.23 kHz, possessing the higher- k components, while the fundamental surface state locates at 5.35 kHz whose field exhibits a Gaussian profile along the direction parallel to the surface. Meanwhile, the hinge states remain to be four degenerate states localized on the hinges, as shown by the field map indicated by the dark blue dot (here, only one hinge state is presented).